# Disturbance Triggers Non-Linear Microbe-Environment Feedbacks

Aditi Sengupta[1†], Sarah J. Fansler[2†], Rosalie K. Chu[3], Robert E. Danczak[2], Vanessa A. Garayburu-Caruso[2], Lupita Renteria[2], Hyun-Seob Song[4], Jason Toyoda[3], Jacqueline Wells[5], and James C. Stegen[2*]

[†]Denotes equal contribution

[1]California Lutheran University, Thousand Oaks, CA

[2]Pacific Northwest National Laboratory, Ecosystem Science Team, Richland, WA

[3]Environmental Molecular Sciences Laboratory, Richland, WA

[4]University of Nebraska-Lincoln, Lincoln, NE

[5]Oregon State University, Corvallis, OR

*Correspondence to*: James Stegen (James.Stegen@pnnl.gov)

## Abstract

Conceptual frameworks linking microbial community membership, properties, and processes with the environment and emergent function have been proposed but remain untested. Here we refine and test a recent conceptual framework using hyporheic zone sediments exposed to wetting/drying transitions. Our refined framework includes relationships between cumulative properties of a microbial community (*e.g.* microbial membership, community assembly properties, and biogeochemical rates), environmental features (*e.g.* organic matter thermodynamics), and emergent ecosystem function. Our primary aim was to evaluate the hypothesized relationships that comprise the conceptual framework and contrast outcomes from the whole and putatively active bacterial and archaeal communities. Throughout the system we found threshold-like responses to the duration of desiccation. Membership of the putatively active bacterial community--but not the whole bacterial and archaeal community--responded due to enhanced deterministic selection (an emergent community property). Concurrently, the thermodynamic properties of organic matter (OM) became less favorable for oxidation (an environmental component) and respiration decreased (a microbial process). While these responses were step functions of desiccation, we found that in deterministically assembled active communities, respiration was lower and thermodynamic properties of OM were less favorable. Placing the results in context of our conceptual framework points to previously

unrecognized internal feedbacks that are initiated by disturbance, mediated by thermodynamics, and that
cause the impacts of disturbance to be dependent on the history of disturbance.

# 1 Introduction

1.1 Conceptual Foundations
Given the influence of microbes over ecosystem function, deeper knowledge of microbe-environment
relationships is needed to improve ecosystem models (Bier et al., 2015). In turn, there is strong interest
in quantifying and predicting microbe-environment relationships such as defining microbial life history
strategies as traits in ecosystem models (Malik et al., 2020), assessing microbial biomass stoichiometry
distributions in response to changing resource environments (Manzella et al., 2019), and evaluating the
extent of microbial adaptation to changing environments and their role in biogeochemical processes
(Wallenstein and Hall, 2012). To enhance and synthesize understanding of microbe-environment
interactions, it is useful to develop conceptual frameworks based on linkages among microbial
characteristics and ecosystem processes. Previous work has used such frameworks to improve
mechanistic representation and predictive capacity of microbe-environment interactions in ecosystem
models (Wieder et al., 2015).
A recently developed framework by Hall et al. (2018) poses a series of concepts that collectively define
the intersection between microbial and ecosystem ecology. Their framework draws attention to causal
relationships between microbial characteristics (microbial membership influences community properties
and microbial processes), which in turn regulate ecosystem fluxes. These components can be further
modified by environmental variation leading to cumulative ecosystems processes with the potential to
incorporate relevant mechanistic links into predictive ecosystem models. Hall et al. (2018) particularly
draws attention to the need to understand how microorganisms influence their environment by
separating microbial community properties based on community aggregated traits (those which can be
predicted using constituent taxa) and emergent properties (properties unable to be explained by
constituent taxa). A powerful element of the framework is that it applies to diverse systems spanning
natural (Gilbert et al., 2018), host-associated (Lloyd-Price et al., 2019), and built (Fu et al., 2020)
environments as well as across spatiotemporal scales (König et al., 2018). While potentially very useful,
the Hall et al. ( 2018) framework has seen little direct use in terms of explicitly defining and evaluating
the linkages within specific study systems (but see Manzella et al., 2019). To make full use of and
continually improve the framework, it is necessary to consider different realizations and interpretations
of the proposed linkages. In the following paragraphs we detail a modified interpretation of the
framework (**Fig. 1**) to enable its application to microbial communities and biogeochemistry associated
with hyporheic zone sediments experiencing hydrological disturbance. In turn, we use data from a
controlled laboratory experiment to evaluate key linkages within the modified framework.

Our modified framework aims to refine the link between microbial communities and ecosystem
functions. As detailed below, a couple critical elements of our modification include:
Defining and evaluating the relative influence of community assembly processes as an emergent
community property, and ii) proposing bi-directional links between environment and microbial
processes to indicate that environmental conditions drive microbial processes, which in turn influence
the environment, and these cumulative environment-microbial processes ultimately drive ecosystem
function.

1.2 Conceptual Framework Development
As in Hall et al. ( 2018) we consider microbial membership to be directly influenced by environmental
conditions (arrow 4, Fig.1) and to underlie community-level properties (arrow 1, Fig. 1). Determining
microbial membership is relatively straightforward, and uses culture-independent (Behrens et al., 2012;
Norland et al., 1995; Thompson et al., 2017; Wagner, 2009) and culture-dependent (Bartelme et al.,
2020) techniques. Sequence-based assays using phylogenetic markers are routine, with DNA-based
(total community members) and RNA-based (putatively active community members) (Barnard et al.,
2015; Blazewicz et al., 2013; Cardoso et al., 2017; Kearns et al., 2016; Shu et al., 2019; Wisnoski et al.,
2020) approaches providing the foundation to study community properties.

While community membership is relatively straightforward, the identification of community properties
that are relevant to a given system and function is open to broader interpretation. Here we propose using
the relative influences of deterministic and stochastic community assembly processes (Stegen et al.,
2012) as emergent properties of microbial communities that have implications for biogeochemical
function (Graham and Stegen, 2017) including in ecosystems experiencing environmental disturbance.
Deterministic mechanisms are associated with systematic differences in reproductive success imposed
by the biotic and/or abiotic environment, while stochastic mechanisms are associated with passive
spatial movements of organisms and birth/death events that are not due to systematic differences across
taxa in reproductive success (Dini-Andreote et al., 2015; Stegen et al., 2015). The relative contributions
of determinism and stochasticity can be inferred by coupling microbial community membership and
phylogeny to ecological null models (Stegen et al., 2012, 2013, 2015; Zhou and Ning, 2017).

We propose that the relative contributions of determinism and stochasticity are emergent properties that
are greater than the sum of the individual components (i.e., taxa), and that are complementary to the
community properties proposed by Hall et al. (Hall et al., 2018), such as biomass and gene expression.
Furthermore, we propose that there are feedbacks between assembly processes and microbial
membership whereby assembly processes influence which taxa are found in which abundances, but
biotic interactions also influence assembly processes. In turn, we modified the framework whereby
arrow 1 is bidirectional (Fig. 1).

It is important to recognize that the ecological processes of community assembly are distinct from
'microbial processes' associated with biogeochemical reactions. As an emergent property, the relative
influence of determinism and stochasticity is the result of complex biotic and abiotic interactions (Grilli
et al., 2017) and also shapes cumulative microbial processes that impact ecosystem biogeochemical
functions (Graham and Stegen, 2017) (arrow 2, Fig.1). For example, a stronger influence of determinism
over community assembly is hypothesized to cause higher respiration rates (a microbial processes) due
to a larger contribution of well-adapted taxa (Graham and Stegen, 2017), though the respiration response
may vary depending on the existing community composition and the deterministic forces exerted.

Analyses of microbial community assembly have been widely employed across environments including
soil (Bottos et al., 2018; Dini-Andreote et al., 2015; Feng et al., 2018; Jurburg et al., 2017; Sengupta et
al., 2019b), sediment (Graham et al., 2017a; Stegen et al., 2013, 2016, 2018b), marine (Starnawski et al.,
2017; Wu et al., 2018), riverine (Chen et al., 2019), gut (Martínez et al., 2015), and engineered (Ofiţeru
et al., 2010; Zhou et al., 2013) systems. Previous work has focused primarily on using DNA-derived
membership and phylogenetic data to study whole-community assembly. In contrast, recent studies have
also used an RNA-based approach to study the relative influence of stochasticity over the assembly of
the putatively active portion of microbial communities (Jia et al., 2020; Jurburg et al., 2017). This RNA-
based approach is complementary to the DNA-based approach and may provide additional insights into
shorter-term dynamic linkages between emergent community properties and microbial processes. Such
linkages have not, however, been previously evaluated.

The Hall et al. ( 2018) framework proposes that microbial processes (e.g., respiration rate) are
influenced by both microbial community properties and environmental factors. Here we propose a
revision of this structure that includes bidirectional links between the environment and microbial
processes (arrow 3, Fig. 1). Such bi-directional links between microbes and their environment are
common (Daly et al., 2016; Leventhal et al., 2019; Ratzke et al., 2018; Stegen et al., 2018a), and in
hyporheic zone sediments may be particularly tied to thermodynamic properties of organic matter
(Graham et al., 2018) and influenced by hydrological disturbances that are common in such
environments. For example, preferential use of OM by microbial communities has potential to alter the
thermodynamic properties of organic matter pools  (Graham et al., 2017a). This microbe-driven shift in
the environment could then feedback to impact microbial metabolism due to the strong influence of OM
thermodynamics on biogeochemical rates (Boye et al., 2017; Garayburu-Caruso et al., 2020; Song et al.,
2020; Stegen et al., 2018b). Bi-directional feedback between environmental factors and microbial
processes is, therefore, likely important to the link between microbial communities and ecosystem
function. Fundamental knowledge of these feedbacks and how they are modulated by hydrologic
disturbances in hyporheic zone sediments is largely unknown, however.

Within ecosystems, mechanistic associations between environmental factors, microbial properties, and
microbial processes underlie spatial and temporal distributions of biogeochemical rates (Fig. 1 arrow 5).
The resulting distributions (e.g., of respiration rates) define cumulative system function and can be used
to understand key phenomena such as biogeochemical hot spots and hot moments (McClain et al.,
2003).  Developing concepts and models to predict the influences of biogeochemical hot spots/moments
is a major outstanding challenge. To facilitate progress, Bernhardt et al. ( 2017) proposed grouping hot
spots/moments into the concept of ecosystem control points that exert a disproportionate influence on
ecosystem function.

While not called out explicitly in Bernhardt et al. (2017), the control point concept is based on the
distribution of biogeochemical rates through space and/or time. Focusing on the shape of rate
distributions allows the notion of control points to be extended to the concept of control point influence
(CPI; Fig. 1 arrow 5). The CPI is a quantitative measurement of the contribution of elevated
biogeochemical rates in space and/or time to the net aggregated rate within a defined system (Arora et
al., 2020). While proposed conceptually and studied via simulation in Arora et al. (2020), empirical
measurements of CPI are lacking. More generally, incorporating CPI into a modified version of the Hall
et al. ( 2018) framework (Fig. 1), provides an integrated conceptualization for how environmental
factors, microbial properties, and microbial processes contribute to emergent system function.

Some elements of our modified framework (Fig. 1) are generalizable across systems (e.g., CPI), while
others (e.g., OM thermodynamics) may have different levels of relevance across different ecosystem
types. Here we aim to generate fundamental knowledge of the linkages between microbial community
and ecosystem function, as well as reveal how hydrologic disturbance may modify these linkages. While
relationships between microbial community assembly and function have been evaluated, integrating the
concepts of microbial structure, function, assembly, and environment interactions into one coherent
framework has the potential to advance our understanding of the feedbacks between microbial
communities and the environment.

1.3 Study Objectives
Our primary objective was to study the modified conceptual framework in context of variably inundated
hyporheic zone sediments exposed to different drying/wetting dynamics. Hyporheic zones are
biogeochemically active subsurface domains in river corridors through which surface water flows and
can mix with groundwater (Bernhardt et al., 2017; Boano et al., 2014; McClain et al., 2003). These
zones can have disproportionate biogeochemical impacts on river corridors (Boano et al., 2014; Burrows
et al., 2017; Demars, 2019; Fischer et al., 2005; Kaufman et al., 2017). Within variably inundated
streams (Larned et al., 2010; Romaní et al., 2006), hyporheic zones experience extreme changes in
environmental conditions, but the consequences of this variability for microbe-ecosystem linkages is
poorly known.

To mimic natural disturbances, we subjected sediments to wetting-drying transitions and focused on a
series of analyses tied to our modified framework. We specifically evaluated relationships between: (*i*)
the relative influence of stochastic assembly as a community property and respiration rates as a
microbial process (Fig. 1 arrow 2), and (*ii*) environmental features and both microbial properties (Fig. 1
arrow 4) and processes (Fig. 1 arrow 3) that underlie aggregate system function (Fig. 1 arrow 5). We
evaluated relationships between cumulative properties of the microbial community (membership,
assembly, biogeochemical rates), environmental features, and emergent ecosystem function by
hypothesizing that: (*i*) Stronger influences of determinism result in well-adapted microbes that will
generate higher respiration rates, (*ii*) Longer duration in an inundated state will result in greater
influences of stochastic assembly--due to weaker ecological selection--and lower respiration rates
following re-inundation due to relatively consistent abiotic conditions (Birch, 1964), (*iii*) Microbial
processes are facilitated by OM that is thermodynamically more favorable for oxidation, leading to an
association between respiration rates and OM thermodynamics, and (*iv*) More wet/dry transitions will
increase among-replicate heterogeneity (e.g., in microbial membership) thereby increasing CPI by
increasing within-treatment variability in respiration rates.

# 194    2 Methods

## 195    2.1 Study site and sediment collection

Hyporheic sediments were collected from the Columbia River shoreline (approximately 46.372411°N,
119.271695°W) in eastern Washington State (Fig S1) (Arntzen, 2006; Goldman et al., 2017; Graham et
al., 2016; Slater et al., 2010; Stegen et al., 2018b; Zachara et al., 2013) within the Hanford Site on
January 14, 2019 at 9 am Pacific Standard Time. Samples were aseptically collected to a depth of 10 cm
at five sub-sampling locations within a meter to form a composite sample that was sieved on site
through a 2 mm sieve into a clean glass beaker. Sieved sediment was stored on blue ice for 30 min while
being transported back to the laboratory. Once back at the laboratory sediment was stored at 4 °C until
processing into incubation vials (see below).

The sediments were subjected to increasing temporal environmental variance (as a function of periodic
wetting and drying transitions) and evaluated for associations between microbial membership, microbial
properties, microbial community assembly, OM chemistry, absolute respiration rate (represented in this
study as $O_2$ consumption rates), and cumulative respiration rates represented as CPI. Aerobic respiration
was chosen as the biogeochemical process since it influences global-scale energy and material fluxes
(Fatichi et al., 2019), and because the hyporheic zone within the field system is predominantly aerobic
(Graham et al., 2016, 2017b). Detailed experimental design and methods are provided in the following
paragraphs.

## 214  2.2 Experimental design

Sediments used in the batch reactors were sourced from one homogenized sediment pool. In turn, 10 g
of sediment from the homogenized pool was added to each reactor vial. The sample set was then split
into two groups, one inundated and the other allowed to desiccate. Desiccation was achieved within 15-
17 days. Sediments were periodically weighed to allow the inundated samples to be maintained at a
constant water content and to monitor the remaining samples for desiccation. Inundated sediments were
placed on ceramic porous plates saturated with DI water for weight adjustments. These conditions were
maintained for 23 days prior to the start of dynamic moisture manipulation, from January 29th to
February 21st, 2019. This initial 'preconditioning' period was used to avoid measuring the immediate
impacts of sampling disturbance and to allow time for desiccation. All replicate reactors were
maintained in the dark, shaking at 100 rpm, at 21 ºC and were covered with a gas permeable Breathe-
Easy (Milli-Pore Sigma, Burlington, MA) membrane that allowed for gas exchange and drying. After
the preconditioning period in which sediments were consistently inundated or allowed to continuously
desiccate, the transition regimes (**Fig. 2**) were applied to the reactors starting on February 22nd, 2019.
We refer to the time from Feb 22nd, 2019 onwards as the 'transitions period' for all treatments, even
though some did not experience transitions between being inundated and dry. Each treatment had 6-7
replicates (detailed below). These regimes were designed around the number of wet/dry transitions
experienced by sediments within a given treatment. Treatment regimes also caused variation in the
cumulative number of days sediments were in a drying state. We imposed six different experimental
treatment regimes (**Fig. 2**) as follows:

●   0 Transitions and 0 days of desiccation: Sediments were maintained at field moisture conditions

for the preconditioning and transitions periods. This treatment had 6 replicates.

●   1 Transition and 34 days of desiccation: Sediments were dry during the preconditioning and

transitions periods, and transitioned once to the field moisture level prior to respiration

estimation. This treatment had 6 replicates.

●   2 Transitions and 4 days of desiccation: Sediments were held at field moisture levels for the

preconditioning period and then for the first 7 days of the transitions period, then transitioned to

a dried state for 4 days, and transitioned to field moisture conditions prior to respiration

estimation. This treatment had 7 replicates.

●   3 Transitions and 31 days of desiccation: Sediments were dry during the preconditioning period

and the first 4 days of the transitions period, then transitioned to field moisture levels for three

245           days, then transitioned to 4 days in a dried state, and transitioned again to field moisture levels

prior to respiration estimation. This treatment had 7 replicates.

●   4 Transitions and 8 days of desiccation: Sediments were held at field moisture levels for the

preconditioning period and transitioned to a dried state for the first 4 days of the transitions

period, then transitioned to field moisture levels for 3 days, then transitioned to 4 days in a dried

state, and transitioned to field moisture levels prior to respiration estimation. This treatment had

7 replicates.

●   5 Transitions and 27 days of desiccation: Sediments were dry during the preconditioning period

and then transitioned to field moisture levels for the first 3 days of the transitions period,

transitioned to a dried state for 2 days, transitioned to field moisture levels for 2 days,

transitioned to a dried state for 2 days, and transitioned to field moisture levels for 3 days prior to
respiration estimation. This treatment had 7 replicates.

To avoid modifying electrical conductivity across experimental treatments, sterile deionized water was
added to reactors to achieve/maintain field moisture levels according to the defined wet/dry regimes
detailed above. For reactors with sediments that were below field moisture levels, deionized water was
added to achieve field moisture levels prior to respiration rate estimation. Changes in the total mass of
reactors and volumes of water added during the course of the experiment are provided in Table S1.

## 264 2.3 Respiration rate measurements

Laboratory incubations were performed in batch reactors to quantify dissolved oxygen consumption
rates. Borosilicate glass vials (20 ml)  (I-Chem™ Clear VOA Glass Vials, Thermo-Fisher, Waltham,
MA)  served as incubator reactors. Factory calibrated oxygen sensor spots (Part# 200001875, diameter =
0.5 cm, detection limit 15 ppb, 0 – 100 % oxygen; PreSens GmbH, Regensburg, Germany) were adhered
to the inner vials of the reactor prior to sediment addition. Detailed description of sensor adhesion and
non-destructive measurements of DO consumption using these sensors is provided in Garayburo-Caruso
et al. (Garayburu-Caruso et al., 2020). Vials were grouped into six treatment regimes (explained in the
previous section) representing inundation-drought transitions.

Sample processing and incubations were performed in a laboratory at 21±1 °C. The reactors were
monitored for 2 hours, with measurement of dissolved oxygen (DO) concentration ($\mu$mol L$^{-1}$) every 30
min.  DO concentration in each bioreactor was measured with an oxygen optical meter (Fibox 3;
PreSens GmbH) connected to a 2 mm polymer optical fiber lined up to sense the sensor dot every thirty
minutes. A few samples were discarded due to sensor dots detaching from the glass surface. Respiration
rates ($\mu$mol L$^{-1}$ h$^{-1}$) were estimated as the slope of the linear regression between DO concentration and
incubation time for each sample. Some non-linearity was observed in the relationship between DO
concentration and time such that only the first 4 data points--time zero to 2 hours--were used to fit a
linear function. The slope of the linear function was taken as an estimate of respiration rate.

## 284 2.4 Microbial Analysis

Post-incubation, the sediment slurry was transferred to centrifuge tubes (Item#28-108 Genesee
Scientific) and centrifuged for 5 min at 3200 rcf and 20°C. The supernatant was removed and reserved
for biogeochemistry analyses and sediment aliquots for DNA and RNA extraction were flash-frozen in
liquid N$_2$ and stored at −80 °C. The extraction, purification, and sequencing of sediment microbial
gDNA were performed according to published protocol (Bottos et al., 2018). The extraction of RNA was
performed using the Qiagen PowerSoil RNA extraction kit (Qiagen, Germantown, MD). RNA was
treated with DNase and quantified with a Qubit RNA kit (Thermo Fisher, Waltham, MA). An aliquot of
the RNA extraction was used to generate cDNA using  SuperScript™ IV First-Strand Synthesis System
(Thermo Fisher Scientific, Waltham, MA). The 16S rRNA gene sequencing--for both gDNA and
cDNA--followed the established protocol by The Earth Microbiome Project (Caporaso, 2018). Sequence
pre-processing, operational taxonomic unit (OTU) assignment, and phylogenetic tree building were
performed using an in-house pipeline, HUNDO (Brown et al., 2018). Sequences were deposited at
NCBI's Sequence Read Archive PRJNA641165. The final sample count of gDNA and cDNA,
respectively, for each treatment regime, after dropping samples following quality filtering and
rarefaction, was 5 and 5 (0 transition), 4 and 3 (1 transition), 5 and 4 (2 transitions), 7 and 6 (3
transitions), 7 and 6 (4 transitions), and 7 and 5 (5 transitions). Rarefaction levels are provided below in
the Statistics section.

## 303   2.5 Biogeochemistry

Reserved supernatant was filtered through a 0.22 μm polyethersulfone membrane filter (Millipore
Sterivex) and an aliquot was immediately removed for non-purgeable organic carbon (NPOC) and the
remainder was stored at -20C until further OM high resolution analysis was conducted (see below).
NPOC was determined by acidifying an aliquot of sample with 15% by volume of 2N ultra-pure HCL
(Optima grade, Fisher#A466-500). The acidified sample was sparged with carrier gas (zero air, Oxarc#
X32070) for 5 minutes to remove the inorganic carbon component.  The sparged sample was then
injected into the TOC-L furnace of the Shimadzu combustion carbon analyzer TOC-L CSH/CSN E100V
with ASI-L auto sampler at 680°C using 150 uL injection volumes.  The best 4 out of 5 injections
replicates were averaged to get the final result.  The NPOC standard was made from potassium
hydrogen phthalate solid (Nacalia Tesque, lot M7M4380).  The calibration range was 0 to 70 ppm
NPOC as carbon.
Fourier Transform Ion Cyclotron Resonance Mass Spectrometry (FTICR-MS) of post-incubation
sediment slurry was conducted as per Danczak et al. (2020). Sample processing, injection, and data
acquisition, processing and analysis was performed as per scripts provided in Danczak et al. (2020), with
'*Start tolerance*' in Formularity changed to 8. Ten samples were dropped due to poor calibration,
resulting in 5 replicates for 0 transition, 4 replicates for 1 transition, 5 replicates for 2 transitions, 6
replicates for 3 transitions, and 5 replicates each for 4 and 5 transition regimes.

From the FTICR-MS data, as in previous work (Garayburu-Caruso et al., 2020; Graham et al., 2018;
Sengupta et al., 2019b; Stegen et al., 2018b), we followed LaRowe and Van Cappellen (LaRowe and
Van Cappellen, 2011) to calculate the Gibbs free energy for the half reaction of organic carbon
oxidation under standard conditions ($\Delta G^0_{Cox}$). This calculation is based on elemental stoichiometries
associated with molecular formulae assigned to individual molecules observed in the FTICR-MS data.
The formulae assignments are part of the processing scripts described in (Danczak et al., 2020). As in
previous work (Garayburu-Caruso et al., 2020; Graham et al., 2018; Sengupta et al., 2019b; Stegen et
al., 2018b), we interpret larger values of $\Delta G^0_{Cox}$ to indicate OM that is thermodynamically less favorable
for oxidation by microbes. That is, larger values of $\Delta G^0_{Cox}$ indicate OM that provides less net energy to a
microbial cell per oxidation event, assuming all else is equal. Given the large numbers of assigned
formulae within each sample, this resulted in thousands of $\Delta G^0_{Cox}$ estimates within each sample, from
which we estimated mean $\Delta G^0_{Cox}$ for each sample.

## 2.6 Estimating influences of community assembly processes

The relative influences of community assembly processes impacting microbial community membership
are emergent properties that cannot be calculated/inferred directly from knowledge of membership. To
evaluate assembly processes as a link between membership and microbial processes (refer Fig. 1) it is
necessary to quantitatively estimate the relative influences of these processes. To do so we use a well-
established null modeling framework based on phylogenetic relationships among microbial taxa  (Dini-
Andreote et al., 2015; Stegen et al., 2012, 2015; Zhou and Ning, 2017). We refer the reader to these
previous studies for details.  In brief, randomizations were used to generate estimates of phylogenetic
associations among microbial taxa for scenarios in which microbial communities were stochastically
assembled. These stochastic (i.e., null) expectations were compared quantitatively to observed
phylogenetic associations to estimate the $\beta$-Nearest Taxon Index ($\beta$NTI) (Stegen et al., 2012).  We used
cDNA sequences rarefied to 27227 and gDNA sequences rarefied to 15106 sequences per sample to
determine putatively active community and whole community $\beta$NTI values, respectively. Samples
falling below these sequence counts were removed as indicated above in subsection *2.4 Microbial*
*Analysis.* A $\beta$NTI value of 0 indicates no deviation between the stochastic expectation and the observed
phylogenetic associations, thereby indicating the dominance of stochastic assembly processes. As $\beta$NTI
deviates further from 0, there is an increasing influence of deterministic assembly processes that drive
community membership away from the stochastic expectation. $\beta$NTI values below -2 or above +2
indicate statistical significance, with negative and positive values indicating less than or more than
expected shifts in membership. $\beta$NTI is a pairwise metric measured between any two
communities/samples, such that shifts in membership are related to changes between the pair of
communities being evaluated. We used $\beta$NTI to study all pairwise community comparisons within each
experimental treatment. Each community from a given reactor is therefore associated with multiple
$\beta$NTI values due to being compared to communities associated with other replicate reactors. In turn, the
average $\beta$NTI was calculated for each reactor. As in Stegen et al. (Stegen et al., 2015), this provides a
community-specific value for $\beta$NTI and thus an estimate of the relative influences of stochastic and
deterministic processes causing deviations between a given community and all other communities within
the same experimental conditions (Sengupta et al., 2019b). That is, the larger the absolute value of $\beta$NTI
for a given community, the stronger the influence of deterministic assembly processes acting on that
community (Stegen et al., 2015). In turn, these community-specific estimates were related to reactor-
specific measurements. For example, respiration rates were regressed against $\beta$NTI to evaluate the link
between emergent properties (i.e., ecological assembly) and microbial processes (i.e., respiration rates).

## 2.7 Evaluating relationships between microbial characteristics and environment

Respiration rate distributions, absolute $\beta$NTI values, and $\Delta G^0_{Cox}$ were summarized as box plots.
Pairwise Mann-Whitney test was performed to evaluate statistical differences between reactor-specific
measurements (*e.g.,*respiration rates and thermodynamic properties) and treatment groups (cumulative
dry and inundated days). Continuous bivariate relationships were evaluated with ordinary least squares
regression. Prior to regression analyses, respiration rates were log-transformed due to non-linearities
resulting from respiration being constrained to be at or above zero. Prior to log-transformation, half the
smallest non-zero rate was added to each rate to enable inclusion of rate estimates with a value of zero.

## 2.8 Control point influence calculation

To characterize respiration rate distributions we used the control point influence (CPI) metric. CPI was
recently developed (Arora et al., 2020) and is defined as the fraction of cumulative function ($R_{tot}$; e.g.,
total respiration rate) within a defined system that is contributed by individual rates that are above the
system's median rate ($R_{med}$). To define cumulative function one must first define the system being
evaluated. In our study, all replicate batch reactors within a given experimental treatment were
conceptualized as a representative set of samples from a larger system experiencing the experimental
conditions. $R_{tot}$ for each treatment was therefore estimated as the sum of respiration rates across a
treatment's replicate reactors. CPI was estimated as the sum of respiration rates that fell above the
median rate for a given treatment ($R_{above}$) divided by $R_{tot}$ for that treatment. That is, $R_{above} = \sum_{i}^{N} R_i$ ,
where $R_i$ are respiration rates from individual reactors that fell above $R_{med}$, and $CPI = R_{above}/R_{tot}$.

An important feature of CPI is that it makes no assumptions of distribution normality, and can be
estimated for rate distributions of any form (e.g., unimodal, multimodal, Gaussian, skewed, etc.). In
most cases, CPI is constrained to have a minimum value of 0.5 (for a perfectly normal distribution with
no outliers) and asymptotically approach 1 as a maximum value (e.g., for heavily skewed distributions
with a small number of very high rates). CPI therefore quantitatively estimates the biogeochemical
contribution of places in space or points in time that have elevated biogeochemical rates.

Understanding how CPI varies through space, time, and with environmental conditions (e.g., disturbance
frequency) provides an opportunity to deepen our understanding of factors controlling hotspot/moment
behavior. Lower values of CPI (i.e., closer to 0.5) can arise through any mechanism that constrains
biogeochemical rates to be consistent through space or time. For example, if respiration rates were
measured in multiple locations across a well-mixed water body, we may expect a Gaussian rate
distribution with little variation. The cumulative respiration of the water body should not be influenced
significantly by hotspots, which would be reflected in low CPI. Conversely, higher values of CPI (i.e.,
closer to 1) can arise through any mechanism that increases the probability of positive outliers within a
given rate distribution. Spatial variation in the temporal dynamics of disturbances (e.g., more frequent
disturbances in some locations) is an example that is closely aligned with our experimental treatments.
In this case the probability of rate outliers (i.e., hotspots) may vary as a function of disturbance
frequency. For example, in spatial domains that experience more wetting/drying dynamics we might
expect a higher probability of rate outliers. This is because each time sediments go dry the exact spatial
distribution of water films that remain will vary across locations within the broader wetted/dried spatial
domain. In turn, we would expect higher values of CPI in spatial domains that are more frequently
wetted and dried. Regardless of whether this specific hypothesis is rejected or not, we contend that using
CPI enables development of *a priori* quantitative hypotheses spanning space, time, environmental
conditions (e.g., disturbance frequency), and scales. This provides new opportunities to more
systematically elucidate factors governing the influence of hotspots/moments.

# 3 Results


To link microbial membership to emergent microbial community properties (Fig. 1 arrow 1) we used
null modeling to estimate the contributions of stochastic and deterministic community assembly. Results
from the null models indicate a relatively balanced mixture of stochasticity and determinism for both the
whole community (gDNA-based) and putatively active community (rRNA-based) (Fig. S2). More
specifically, stochasticity and determinism each governed 50% of turnover in microbial membership for
the whole community and 33% and 67%, respectively, for the putatively active community. The relative
contributions of the two deterministic components (homogeneous and variable selection) were strongly
imbalanced. Homogeneous selection was responsible for 94% and 91% of the deterministic component
for the whole and putatively active communities, respectively. The contributions of homogeneous and
variable selection to the deterministic component must sum to 1, such that the variable selection was
responsible for 6% and 9% of the deterministic component for the whole and putatively active
communities, respectively.

As shown in Figure 1 (arrow 2), we hypothesized a link between microbial community properties and
microbial processes realized as a relationship between the strength of determinism and respiration rates.
Such a relationship was not observed for the whole community (Fig. 3a), but we did observe a non-
linear decreasing relationship between respiration rates and the absolute value of βNTI for the putatively
active community (Fig. 3b). The direction of this relationship (negative) was opposite of that expected,
and the relationship was clearly structured by among-treatment shifts in both respiration rate and βNTI
(Fig. 3b).

In our conceptual framework there are multiple ways in which connections among the environment,
microbial properties, and microbial processes may be realized, in part due to the environment having
multiple components relevant to our study (Fig. 1 arrows 3,4). More specifically, the environment was
characterized here in terms of both disturbance (number of dry days; imposed by the experimental
manipulation) and OM thermodynamics ($\Delta G^0_{Cox}$; this is an emergent aspect of the environment).

Disturbance influenced both microbial properties and processes. These influences appeared to be non-
linear with experimental treatments associated with the two largest number of dry days (31 and 34)
causing decreases in respiration rates (Fig. 4a) and stronger influences of deterministic homogeneous
selection for the putatively active community (Fig. 4b). Disturbance had no clear influence on
community assembly for the whole community (Figs. S3, S4a). Given the apparent binary nature of
these results, we evaluated statistical significance by combining respiration rate data from treatments
with 0-27 cumulative dry days and separately combining data from treatments with 31 or 34 cumulative
dry days (Fig. S5). Respiration rates were significantly depressed in the treatments associated with 31 or
34 cumulative dry days ($W = 5$, $p < 0.001$). The βNTI data are non-independent due to being based on
all pairwise comparisons within a treatment. Standard statistical tools are therefore not applicable for
assigning statistical significance when comparing βNTI distributions. However, as shown in Figures 4b
and S4b, there is an obvious shift to lower βNTI values for the putatively active community in the
treatments with 31 or 34 cumulative dry days.

The other aspect of the environment examined here (i.e., OM thermodynamics) also had significant
relationships with both microbial processes (Fig. 1, arrow 3) and properties (Fig. 1, arrow 4). More
specifically, respiration rates decreased significantly as a negative exponential function of increasing
$\Delta G^0_{Cox}$ ($R^2 = 0.34$; $p = 0.001$; Fig. 5a). This indicates a decrease in respiration rate as OM
thermodynamic properties shifted towards lower favorability for oxidation (i.e., larger values of $\Delta G^0_{Cox}$).
Similarly, we found that the strength of deterministic assembly associated with the putatively active
community increased linearly with $\Delta G^0_{Cox}$ ($R^2 = 0.40$; $p = 0.001$; Fig. 5b). The relationships were clearly
structured by among-treatment shifts in respiration rate, βNTI, and favorability of organic matter
($\Delta G^0_{Cox}$). The strength of deterministic assembly associated with the whole community was unrelated to
$\Delta G^0_{Cox}$ ($p = 0.64$) (Fig. S6).

The conceptual model described in Figure 1 focuses primarily on connections among environmental
and/or microbial attributes, but there are potentially important relationships within attribute categories.
In particular, within the environmental category there is the potential for an influence of disturbance on
OM thermodynamics. Such an effect was found for OM thermodynamics as measured by $\Delta G^0_{Cox}$ (Fig.
5c). Using the same approach as for analyses described above, we combined data for treatments with 0-
27 cumulative dry days and compared that distribution to data combined across treatments with 31 or 34
cumulative dry days. A Mann-Whitney test comparing these distributions confirmed a significant
change in the $\Delta G^0_{Cox}$ distribution (W = 189, p = < 0.001)(Fig. S7).

The last component of the conceptual model considered here is the connection between microbial
processes occurring in a given location and cumulative system function that aggregates across locations
(Fig. 1, arrow 5). It is at the system level that the influence of biogeochemical hot spots (or hot
moments) can be evaluated. We conceptualized an aggregate system as the collection of replicate batch
reactors within a given experimental treatment. Based on this definition, we estimated control point
influence (CPI) as a measurement for the influence of biogeochemical hot spots. We observed a large
amount of variation in CPI across experimental treatments, but there was no clear, direct influence of the
treatments on CPI (Fig. 4). The largest value of CPI observed (> 0.9) was associated with the treatment
that imposed 31 cumulative dry days. This treatment also had the lowest median respiration rates across
all treatments (Fig. 4). We did not find any significant relationship ( p > 0.70) between within-treatment
community compositions (whole and putatively-active) and CPI, suggesting that within-treatment beta-
dispersion of the community is not a better predictor of CPI.

# 491   4 Discussion

Mechanistic evaluation of microbe-environment interactions is fundamental to understanding microbe-
mediated ecosystem function. Inspired by a microbe-environment-ecosystem framework proposed by
Hall et al. (Hall et al., 2018) we proposed and evaluated a modified framework linking microbial
characteristics (membership, emergent properties, processes), the environment (disturbance, OM
thermodynamics), and cumulative ecosystem function (CPI) of hyporheic zone sediments. Our results
provide clear support for the overall conceptual framework and further point to an iterative loop among
OM thermodynamics, respiration rates, and microbial community assembly that can be initiated by
externally-imposed disturbance. Furthermore, our results indicate that the iterative thermodynamics-
assembly-respiration loop may be initiated through threshold-like impacts of disturbance that were
observed only after 31 or more cumulative days of desiccation.

We first evaluated emergent community properties as a function of microbial membership by studying
the relative influences of stochasticity and determinism over community assembly. Taking this
approach, we found fully balanced stochastic-deterministic influences over the whole community, in
which each contributed to 50% of the variation in community composition. The relative influences of
stochasticity and determinism have been quantified for many microbial systems and the estimates are
highly variable (Tripathi et al., 2018; Wang et al., 2013). In addition, within the deterministic
component of assembly, homogeneous selection had a far greater influence than variable selection.
Previous work has also observed a broad range of contributions from homogeneous and variable
selection (Fillinger et al., 2019; Graham et al., 2016; Li et al., 2019; Sengupta et al., 2019a; Whitman et
al., 2018). As such, the assembly-associated outcomes observed here for the whole community are not
unexpected relative to previous work. Very few studies, however, have examined the relative influences
of different assembly components over putatively active microbial communities. This focus is critical in
our framework since many microorganisms are inactive while the active community remains sensitive to
abiotic stresses and contributes to ecosystem function.

For the putatively active communities we found that across all treatments both stochasticity and
determinism were important, though deterministic assembly had greater influence. This deviates
quantitatively from the whole community in which the influences of stochasticity and determinism were
more balanced. The difference in assembly influences in whole- and putatively-active communities is
likely driven by the duration of the imposed experimental treatments (two weeks). This time period may
not be sufficient for birth and death events to restructure the community composition. Instead,
physiological responses as a function of decreases and increases in microbial activity (changes in cDNA
signatures) is more likely. A stronger disturbance (*e.g.,* imposing the treatments for a longer period of
time) may provide further insights into the impacts of disturbance on the whole community (*e.g.,* evident
as changes in gDNA signatures). Consistent with the whole community results, however, was the
dominance of homogeneous selection within the deterministic component of assembly. The strong
influence of homogeneous selection is likely due to selection-based constraints imposed by aspects of
the experimental system that did not vary across treatments. For example, mineralogy is known to
strongly influence microbial communities (Boyd et al., 2007; Carson et al., 2009; Doetterl et al., 2018;
Fauvel et al., 2019; Mauck and Roberts, 2007; Stegen et al., 2016) and was homogenized across the
experimental batch reactors, thereby potentially imposing homogeneous selection on both the whole and
putatively active communities.

Our study uniquely evaluates null-model outcomes of putatively-active community assemblies in
hyporheic zone sediments, where homogeneous selection was further enhanced by our experimentally
imposed hydrologic disturbances. Increased homogeneous selection in response to disturbance is
consistent with previous work in aquatic (Chase, 2007) and soil systems. For example, in a soil system,
disturbance led to an immediate increase in homogeneous selection for the putatively active community
(Jurburg et al., 2017). The strong influence of homogeneous selection on the putatively active
community is not always observed, however, suggesting it may be tied to acute disturbance. That is, Jia
et al. (2020) recently found that within a natural soil chronosequence, variable selection was stronger for
putatively-active communities while homogeneous selection influenced the whole community assembly.
Our results combined with these previous studies indicate that community assembly of putatively-active
members may be more closely linked to short-term environmental change than assembly of the whole
community.

In addition to being more sensitive to disturbance, we find that assembly of the putatively active
community was more strongly tied to microbial processes (i.e., respiration rate), than was the whole
community. A strong link between biogeochemical rates and the putatively active community is
consistent with previous studies (Freedman et al., 2015; Levy-Booth et al., 2019). More specifically, we
observed a negative relationship between respiration rate and absolute values of βNTI for the putatively
active community, but no relationship for the whole community. The direction of this relationship is
opposite to our hypothesis. While stronger selection should remove mal-adapted individuals, leading to
higher biogeochemical rates (Graham and Stegen, 2017), increased selection in our experiment was
imposed by disturbance that appeared to directly suppress respiration rates due to desiccation (Baldwin
and Mitchell, 2000; Manzoni et al., 2012). The simultaneous suppression of respiration and imposition
of stronger selection in treatments with 31 or 34 days of dry conditions (Fig. 3b) led to the negative
relationship between respiration and the strength of selection. The lack of such relationships when
considering the whole community indicates that a greater focus on assembly dynamics of putatively
active communities could reveal new insights into the multi-component linkages among microbes, the
environment, and function.

Disturbance also impacted OM thermodynamics and respiration rates, potentially initiating an iterative
loop among microbial assembly, microbial processes, and the abiotic environment. In this iterative loop
the direction of causation between OM thermodynamics and microbial processes (Fig. 1, arrow 3) is not
clear due to feedbacks, though we interpret a direction of causation from OM thermodynamics to
microbial properties in terms of community assembly (Fig. 1 arrow 4). As such, there may be a loop
between OM thermodynamics and microbial processes (i.e., respiration) embedded in a larger loop that
also includes microbial properties (i.e., community assembly). Such feedbacks are inherent in complex
systems and often lead to non-linear dynamics as observed here in terms of the threshold-like impact of
desiccation on multiple system components (Pérez Castro et al., 2019; Prosser and Martiny, 2020).

As key elements of the inferred system of feedbacks, the links among OM thermodynamic properties,
respiration, and desiccation found here are consistent with recent work tied to the same field system.
That is, Garayburu-Caruso et al. (Garayburu-Caruso et al., 2020) also showed decreasing aerobic
respiration with decreasing favorability for oxidation (i.e., larger values of $\Delta G^0_{Cox}$) using sediments
sourced ~2 years previously from the same field system. In addition, the impacts of desiccation found
here are similar to Goldman et al. (Goldman et al., 2017) after re-inundation. This impact of desiccation
on respiration contrasts with the Birch effect (Birch and Friend, 1956) in soils whereby desiccation
followed by re-wetting leads to enhanced respiration. The consistency across hyporheic zone studies and
deviation from classical soil phenomena points to potential consistency in governing processes within
the hyporheic zone that deviate from processes operating in soil systems. Further evaluation is needed
across additional hyporheic zone systems to rigorously evaluate this inference, however.

In addition to linkages between the environment and microbial aspects of the system, our study revealed
connections within the environmental components of the conceptual framework. That is, greater
cumulative desiccation caused an increase in $\Delta G^0_{Cox}$, indicating a significant change in OM
thermodynamics (Fig. 5c). While our data cannot pinpoint governing mechanism(s), we hypothesize that
the $\Delta G^0_{Cox}$ response may have been tied to increased ion concentration following desiccation. For
example, OM chemistry may have been altered due to changes in abiotic sorption, limitations of
microbially accessible C due to water potential constraints, and/or osmolyte production and formation of
extracellular polymeric substances (Fierer et al., 2003; Gionchetta et al., 2020; Homyak et al., 2018).

Irrespective of mechanisms, the shift in OM thermodynamics in response to desiccation was associated
with a decline in respiration. We infer a causal connection between OM thermodynamics and
respiration, potentially triggered by desiccation-driven shifts in OM chemistry and/or microbial
physiology. This causal connection is supported by recent work (Garayburu-Caruso et al., 2020) and the
observation of a continuous function between $\Delta G^0_{Cox}$ and respiration rate that transcended experimental
treatments. Desiccation therefore likely influences and may even initiate an iterative loop among OM
thermodynamics, microbial assembly, and biogeochemistry that underlies cumulative system function.

Cumulative system function can often be driven by ecosystem control points (Bernhardt et al., 2017),
but we observed relatively little indication of such behavior. That is, estimates of control point influence
(CPI) were relatively low across most treatments. CPI is theoretically constrained to range from 0.5-1,
with lower values indicating smaller influences of control points. In our study, all but one treatment had
CPI between ~0.5 and 0.7. The associated distributions of respiration rates did not contain obvious
outliers such that we interpret CPI values in the 0.5-0.7 range to be relatively low and not strongly
influenced by control points or biogeochemical hot spots/moments (McClain et al., 2003). The treatment
with 31 cumulative days of desiccation diverged from the rest in having a CPI value of ~0.9. This large
CPI was due to a single outlier (Fig. 5a) such that most of the cumulative respiration across reactors was
contributed by that single reactor. We interpret that single reactor as a biogeochemical hot spot or
control point within that experimental treatment. It is unclear, however, what led to such behavior as
disturbance did not have any systematic influence on CPI.

A strength of CPI as a metric is that it allows for direct quantitative comparisons across studies, systems,
and scales. Ours is the first study to estimate CPI, however, such that we cannot yet make comparisons
to previous work. Through future comparisons it will be possible to evaluate the strengths, weaknesses,
and behavior of CPI. We expect that some patterns may emerge such as CPI having a greater likelihood
to reach very high values (near 1) in systems with relatively low rates on average. In these conditions,
even a modest quantitative increase in biogeochemical rates can lead to a large proportional change such
that most cumulative function is from a single point in space and/or time, resulting in large CPI. We also
expect that some biogeochemical processes will show greater variation in CPI than others, potentially
due to variation in degree of functional redundancy (Louca et al., 2018). For example, processes such as
respiration can be performed by numerous microbial taxa (i.e., there is high functional redundancy),
while others are more constrained to a relatively small number of taxa (e.g., ammonia oxidation). We
hypothesize that CPI may be lower on average and less variable across systems and scales for
biogeochemical processes with greater functional redundancy. Additional work will be needed to test
this hypothesis.

# 5 Conclusions

In this study we coupled intrinsic characteristics of natural hyporheic zone sediments with imposed

constraints in the form of desiccation to evaluate an *a priori* conceptual framework modified from Hall

et al. (Hall et al., 2018). Our results demonstrated strong and often non-linear connections among

desiccation, OM thermodynamics, assembly of the putatively active microbial community, and

respiration rates. Collating our results points to further modification of the framework into an *a*

*posteriori* conceptual model containing nested feedback loops (**Fig. 6**). This conceptual model is

consistent with the recently proposed unification of microbial ecology around the concepts of external

forcing, internal dynamics, and historical contingencies (Stegen et al., 2018a). That is, we hypothesize

that external forcing imposed by desiccation initiates multiple internal loops that drive biological and

chemical dynamics that, in turn, underlie respiration responses to re-wetting that are contingent on

desiccation history. The development of conceptual models such as this is key to incorporating

additional mechanistic detail into predictive simulation models (e.g., reactive transport codes). We

encourage further evaluation and improvement of both our *a priori* and *a posteriori* concepts across

environmentally divergent conditions to generate knowledge that is transferable across systems.

**Code availability**

The R scripts used in this study are hosted on ESS-DIVE in the data package found at

https://data.ess-dive.lbl.gov/view/doi:10.15485/1807580. Null modeling scripts were based on

those available at https://github.com/stegen/Stegen_etal_ISME_2013.

**Data availability**

Data are available on the ESS-DIVE archive at the following link https://data.ess-
dive.lbl.gov/view/doi:10.15485/1807580. Sequence data is available on NCBI's Sequence Read
Archive PRJNA641165.

**Supplement**

The supplement related to this article is available online at: XXX

**Author contributions**

JCS and SF conceptualized and designed the study; JT, JW, LR, RC, SF, and VAG-C performed
the experiment; AS, JCS, RED, and SF analyzed data; AS, JCS, and SF drafted the manuscript
and all authors contributed to further writing.

**Competing interests**

The authors declare that they have no conflict of interest.

**Acknowledgements**

We thank Amy Goldman and Nathan Johnson for developing graphics. The initial experimental stages
of this work were supported by the PREMIS Initiative at the Pacific Northwest National Laboratory
(PNNL) with funding from the Laboratory Directed Research and Development Program at PNNL, a
multi-program national laboratory operated by Battelle for the US Department of Energy under Contract
DE-AC05-76RL01830. The later stages of this work (e.g., data analysis, conceptual interpretation,
manuscript development) were supported by the U.S. Department of Energy-BER program, as part of an
Early Career Award to JCS at PNNL. A portion of the research was performed using EMSL, a DOE
Office of Science User Facility sponsored by the Office of Biological and Environmental Research.

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

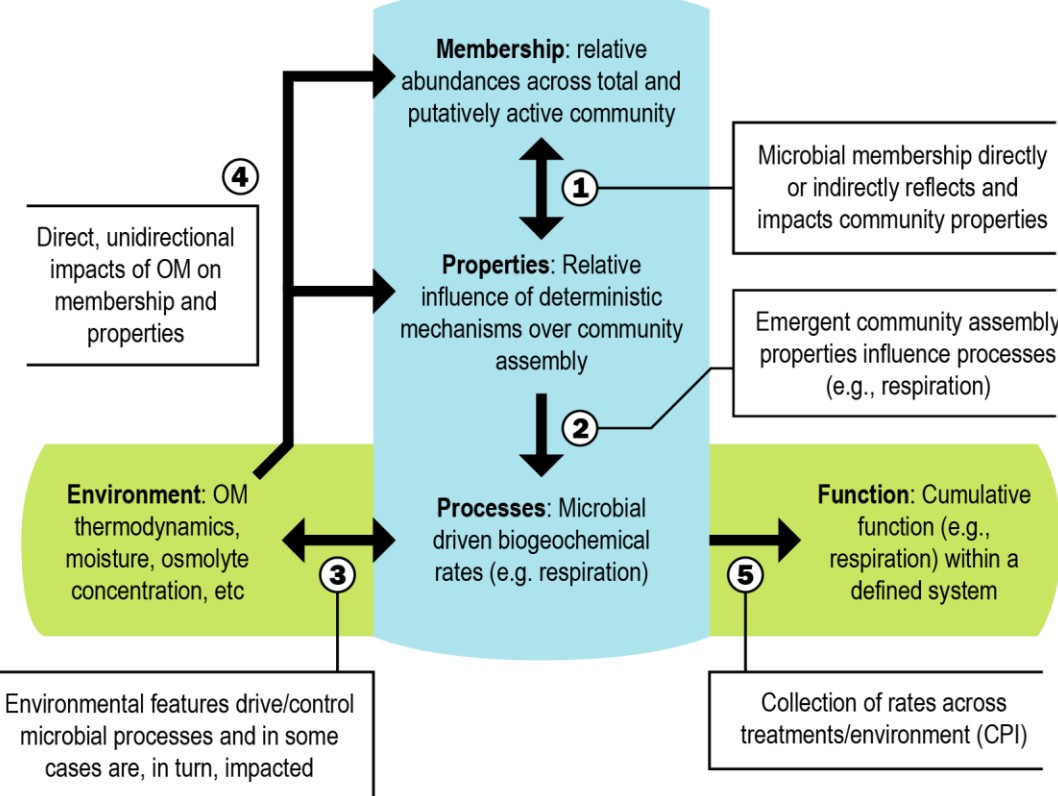

**Figure 1. Integrated conceptual framework.** The conceptual figure (modified from Hall et al. (2018))
details relationships (indicated by numbered arrows) between cumulative properties of the microbial
community (*e.g.*, microbial membership, community assembly properties, biogeochemical rates),
environmental features [*e.g.*, organic matter (OM) thermodynamics], and emergent ecosystem function
[e.g., control point influence (CPI)]. Double headed arrows indicate feedbacks.

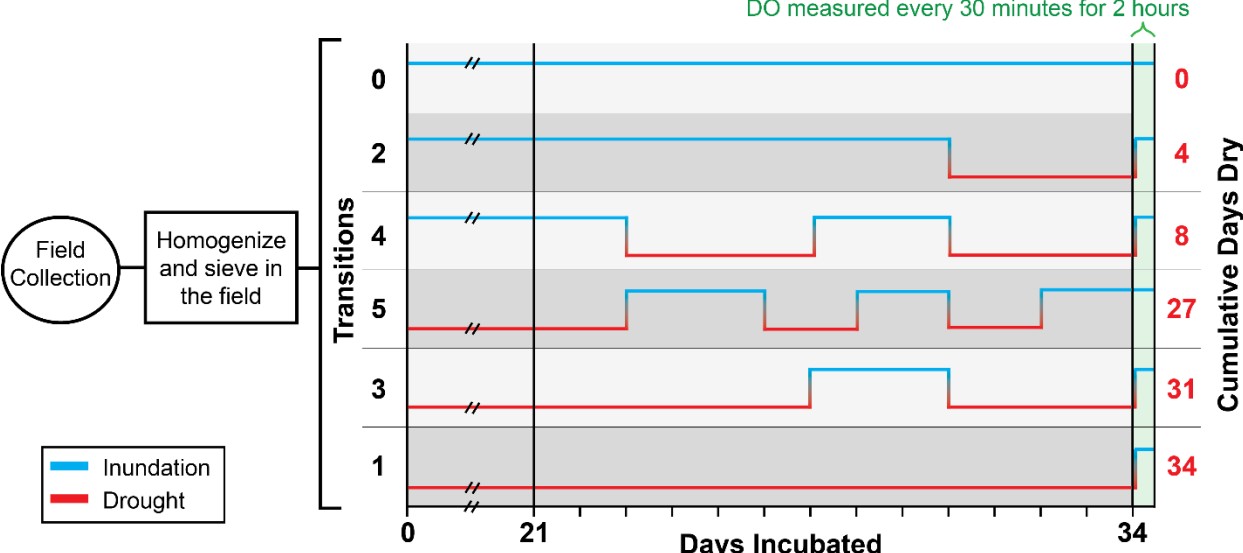

**Figure 2. Experimental design of batch reactor incubations subjected to six treatment regimes**
**of inundated (blue line) and dry (red line) conditions.** Black values on the left indicate the number of
inundated/dry transitions, including the final inundation that all treatments experienced immediately
prior to the measurement of respiration. Red values on the right indicate the number of cumulative dry
days (e.g., treatments with 1 or 3 transitions experienced 34 or 31 cumulative dry days, respectively.
Transitions between inundated and dry conditions started on day 24. All treatments were held at either
an inundated or dry state prior to day 24. Treatments are ordered by the number of days dry.

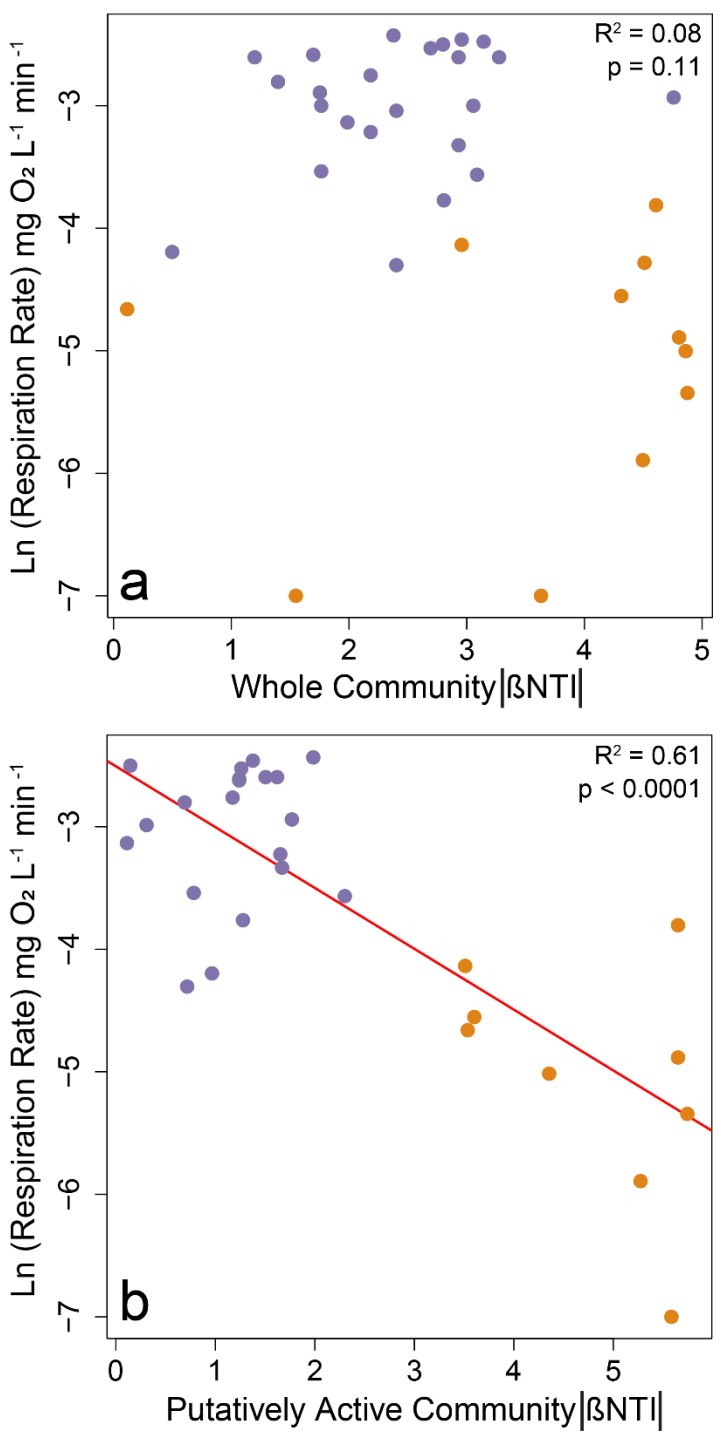

**Figure 3. Natural log transformed respiration rates (i.e., O₂ consumption) as a function of the**
**absolute value of βNTI for (a) whole communities or (b) putatively active communities.** Larger absolute
values of βNTI indicate stronger influences of deterministic assembly. Nonlinearity was observed because the
respiration rate has a lower limit of 0 such that its relationship with βNTI was fit as a negative exponential
function. The significant regression model is shown as a red line, and statistics are provided on each panel.
Treatments with 0, 4, 8, or 27 days dry are shown in purple. Treatments with 31or 34 days dry are shown in
orange.

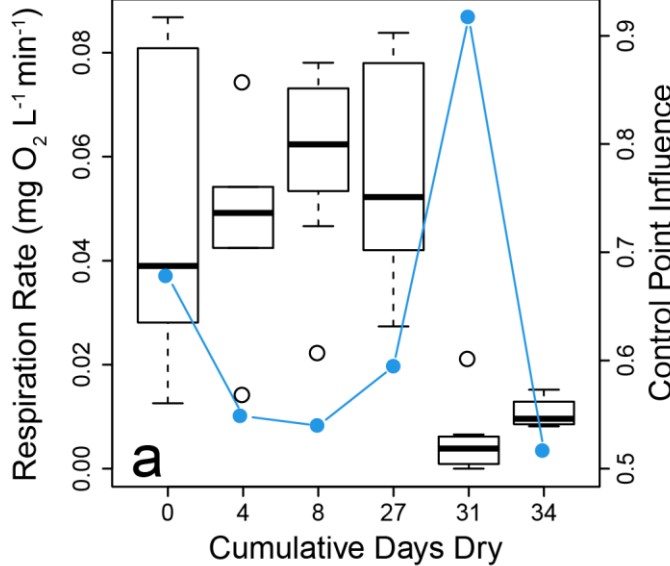

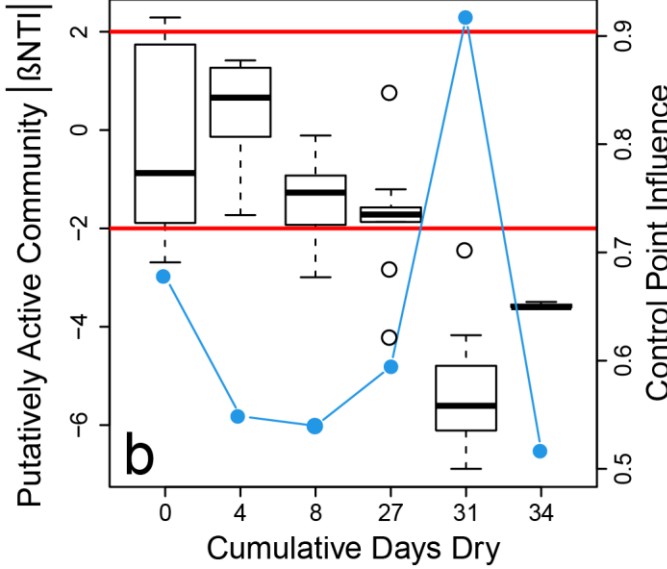

**Figure 4. Boxplot representations of respiration rate (a) and putatively active community βNTI**
**(b) distributions as a function of the cumulative number of days reactors were in a dried state.**
Each value along the horizontal axis represents a different experimental treatment. On both panels the
right hand axis provides estimates of control point influence (blue circles and lines) across the
treatments. Horizontal red lines in (b) indicate significance thresholds; values below -2 indicate
deterministic homogenous selection, values above +2 indicate deterministic variable selection, and
values between -2 and +2 indicate stochastic assembly.

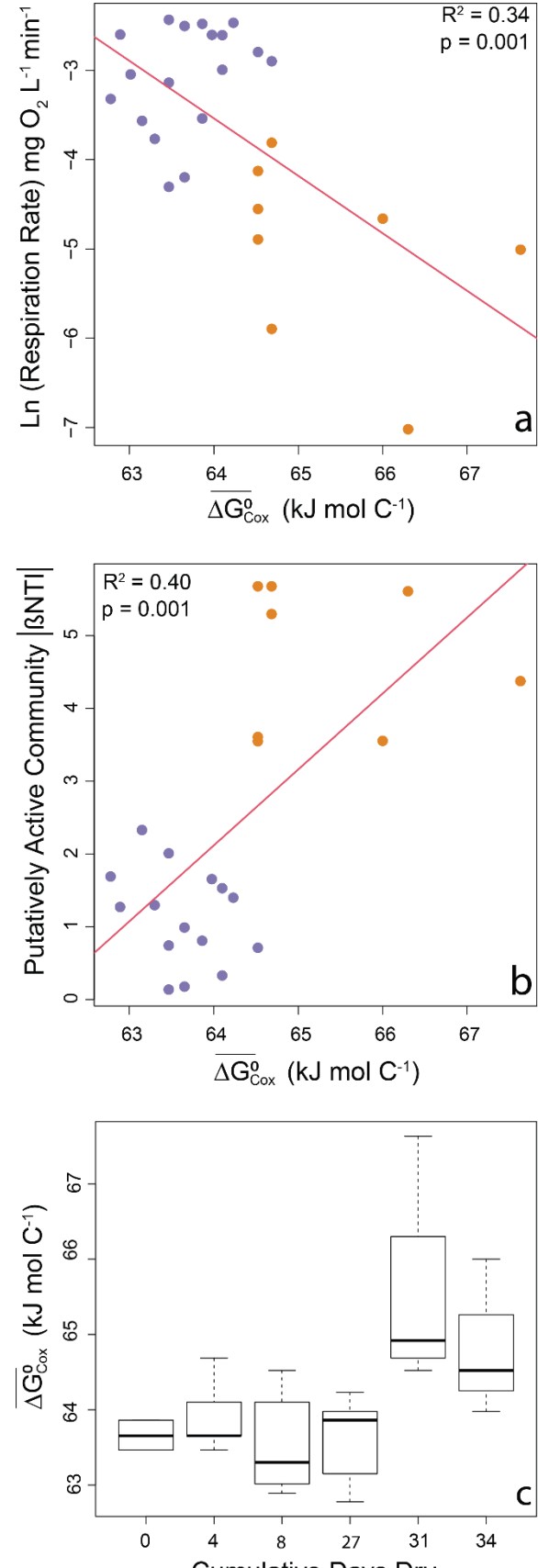

**Figure 5. Microbial processes and properties as a function of OM thermodynamics, and impacts**
**of disturbance on OM thermodynamics.** (a) Respiration rates (natural log transformed) decreased
with decreasing favorability for oxidation (larger values of $\Delta G^0_{Cox}$). (b) The strength of deterministic
selection measured as the absolute value of βNTI increased with decreasing favorability for oxidation. Regression
models are shown as red lines and statistics are provided on each panel. (c) Boxplot representations of the
distributions of OM thermodynamics across experimental treatments. Significant increases were observed for
treatments with 31 or 34 cumulative dry days. See text and Figure S7 for a description of statistics. Treatments
with 0, 4, 8, or 27 days dry are shown in purple. Treatments with 31or 34 days dry are shown in orange.

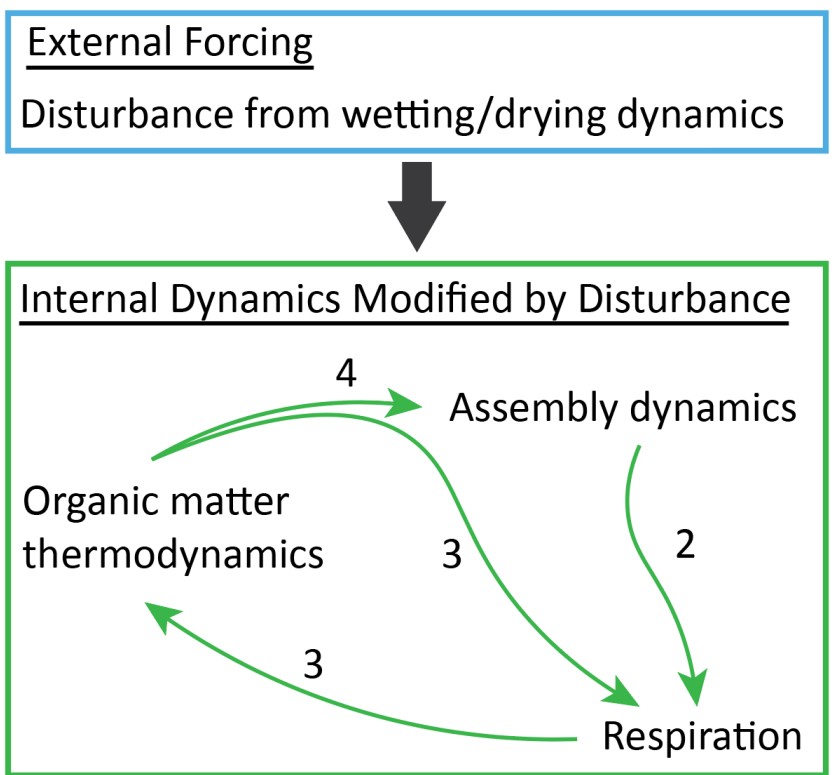

**Figure 6. Integrated conceptual interpretation of results from this study.** Collectively, our results
indicate that the external forcing imposed by disturbance leads to feedback between assembly of the
putatively active community and respiration rates that is modulated by coupled dynamics in organic
matter thermodynamics. Relative to Fig. 1, here external and internal aspects of the environment are
separated. The arrows within the internal dynamics component are analogous to arrows 2,3, and 4 in
Figure 1. The arrow from external to internal is not considered in Figure 1, and represents the impact of
external forcing on all aspects of the internal system. These impacts are both direct effects of
disturbance and indirect effects mediated through the internal feedback that collectively lead to impacts
of re-wetting that are contingent on desiccation history.