# Peer review of "Disturbance Triggers Non-Linear Microbe-Environment Feedbacks"

_Biogeosciences, 2021_

## Author Response (AR1)

We have responded (responses are italicized) to AE and reviewer comments below. The line numbers indicated in our response correspond to the clean revised copy of our manuscript (edits are shown in blue text).

**Comments from Associate Editor:**

1. L. 18: define the abbreviation OM here

*Edited*

2. L. 107: change to use of OM

*Edited*

3. L. 188: change to rpm

*Edited*

4. How quickly did sediments dry out? Just wondering how fast desiccation actually occurred. Also, how was moisture content of the sediment measured?

*Desiccation was achieved within 15-17 days. Sediments were periodically weighed to allow the inundated samples to be maintained at a constant water content and to monitor the remaining samples for desiccation. Inundated sediments were placed on ceramic porous plates saturated with DI water for weight adjustments.*

5. Reminder to omit author name in the parenthetical citation when referring to a paper in a sentence, e.g., as Hall et al. (Hall et al. 2018) — the second Hall et al should be removed

*Edited*

6. L. 276: omit organic carbon after NPOC

*Edited*

7. L. 300-302: influences is used twice in this sentence and makes it awkward to read, consider revising

*Edited*

8. L. 303-304: do you mean to reference Figure 1 here?

*Yes, edited to reflect Figure 1.*

9. Code Availability section states that R scripts are available upon request. But, the SI states that the scripts are available in ESS-DIVE. Please check and modify for clarity as needed.

*Edited. We are in the process of uploading the data to ESS-DIVE and will direct readers to it when data and code is requested.*

10. Figure 1: please define CPI and OM abbreviations in the figure captions.

*Edited*

11. Figure captions are well written.

*Thank you.*

12. Figure 2: I agree with the reviewers that re-ordering the treatments here to match the presentation in the other figures would be helpful

*Edited*

**Reviewer 1** Review Comments (Stephanie Jurberg):

1. The authors present work of good quality, and i especially commend them for making the data available already. My biggest concern is the presentation of the experiment's hypotheses as a conceptual framework, as the notion that microbial community assembly influences and is influenced by microbial functions is not novel. Rather, the authors present what seems to be the basis for a very interesting SEM in their final figure. They might consider pursuing this! Additional comments are listed below in order of decreasing importance.

- *Thanks for your constructive feedback. We have reworked pieces of the introduction and hypothesis in 64-65, 161-165,182-185, and 190-192 to direct readers attention to the need for this relationship to be tested. We would also like to note that while relationships between microbial community assembly and function may not be novel, integrating the concepts by tying in various pieces of our conceptual understanding of microbial structure, function, assembly, and environment interactions into one coherent framework we feel is novel, and in our opinion has value.*

- *Unfortunately we do not have enough data to develop a SEM. We would need at least a 100 data points as pointed out in the book "A Beginners Guide to Structural Equation Modeling" by Randall E. Schumacker. In addition, the relationship between dry days and the response variables is a step function. SEM needs linear relationships, which can sometimes be achieved through transformation, but in the case of a step function it is not clear how to make this linear through transformation. While it would be fantastic to formally test the conceptual model as an SEM, that will need to wait for future studies due to technical limitations.*

2. The authors should be careful with their description and interpretation of deterministic forces throughout the manuscript. Deterministic forces are caused both by biotic interactions and abiotic forces, and the research presented cannot disentangle these two. I therefore think it is important that the authors temper their discussion of determinism as an informative emergent property. In L85, they state "a stronger influence of determinism over community assembly is hypothesized to cause higher respiration rates (a microbial processes) due to a larger contribution of well-adapted taxa (Graham and Stegen, 2017).", however this is entirely dependent on the identity of the deterministic force and the community it is exerted upon. A stronger influence of determinism on microbial community assembly can be caused by an infitnity of factors, but it is these factors that determine how the community is shaped. Thus the expected effect on the community cannot be linked directly to the change in the relative influence of determinism.

- *We agree that our work cannot disentangle biotic vs abiotic forces of deterministic assembly. We have edited the sentence in lines 108-109 to indicate that this may be one possible outcome but may be dependent on the existing community composition and the deterministic force that is exerted.*

3. While I think that the sequencing of both gDNA and cDNA is a strong feature of this manuscript, I don't think that the authors discuss it in sufficient depth. Why might the disturbances change the relative strength of deterministic forces according to cDNA but not gDNA?

- *We hypothesize that the duration of imposed experimental treatments (2 weeks) was not a sufficient amount of time for birth and death events to restructure the composition of the community. The data suggest that instead there were physiological responses in terms of decreases and increases in activity level without strong changes in composition itself. If we would have imposed a stronger disturbance and/or imposed the treatments for a longer period of time, we may have observed clearer impacts of disturbance on the whole community (i.e., via gDNA analyses). We elaborated on this idea in the Discussion in lines 516-518 and 521-527.*

4. In figure 2, the authors present the treatments in a different order than what is used throughout the rest of the manuscript (i.e., dry days). I think this is a great description of a complex experiment, and I would suggest the treatments are reordered to reflect the order in the analyses

- *We have reordered the panels to reflect that in the analyses.*

5. In figures 3 and 5, I would like to see the points colored by treatment. In 5B, the positive correlation is driven (almost?) entirely by the differences between the two treatments with most dry days and the rest. I would be curious to see whether that is the case in the rest of the figures.

● We will do this and provide discussion on any new inferences that emerge.

*We edited the figures to represent the point colored by Cumulative dry days for both Figures 3 and 5. In Figure 3b, the relationship was clearly structured by differences in between the dry and inundated treatments (lines 434-436, 559). Similarly, relationships in Figure 5a (respiration rate and favorability of organic matter) and 5b ( βNTI of putatively active community and favorability of organic matter) were driven by differences in the dry and inundated treatments (Lines 464-466).*

**Reviewer 2**

General comments:

1. This is an interesting paper and I think it represents an important advance in linking microbial community structure with function by using an experiment to support a revised conceptual framework. In this study, Sengupta et al. find that beyond a desiccation threshold, microbial communities experience strong homogeneous selection for a subset of taxa and respiration rates plummet. Interestingly, stronger deterministic selection (likely due to strong environmental filtering) is associated with reduced respiration (again likely due to strong environmental controls on this function), with a transition towards less favorable OM thermodynamic conditions as a putative mechanism connecting the community with the function. I enjoyed reading the study and offer more detailed suggestions below to increase the clarity of the manuscript.

● *Thank you for your feedback and comments. We have addressed the comments in detail in our revision.*

2. I know there is probably a word limit for the abstract, but having read it before reading the full manuscript, I was unclear about the study's aims and how to interpret the results from the study. I think there are some really interesting results that perhaps could be more clearly articulated in the abstract.

● *We edited the abstract to improve the messaging (line 14-19 and 24-26).*

3. The introduction is somewhat long (11 paragraphs). As a reader, I was losing sight of the big picture. In particular, one major contribution of this paper is a modification of the Hall et al. framework, which I think is somewhat buried in the introduction. What about making an additional section (starting on Line 53) called "Refining the link between microbial communities and ecosystem functions" or something that would better emphasize the conceptual contributions of this manuscript?

- *We have edited the introduction (by creating sub-sections) per the reviewer's suggestion to help make it more succinct and highlight conceptual contributions of this manuscript (line 46-48, 64-65, 97-100). We have also summarized the main findings of the Hall et al., study (lines 49-53) as commented as a line-specific comment below.*

Specific comments:

4. Lines 11-14: Can the essence of the conceptual framework be communicated more specifically? Perhaps emphasize the causal linkages (from Fig. 1) so the reader will understand the context of the results presented in the abstract (see also line 21)?
- *Edited (lines 14-19)*

5. Line 16: Maybe specify that these are bacterial communities
- *Edited*

6. Lines 18-20: I understand what this means after looking at figures 3 and 4, but maybe "relationships among community assembly, respiration, and OM thermodynamics" could be rephrased slightly so it doesn't appear to conflict with the description of respiration as a step function (line 18, referring to transitions with duration of drying). What about something like "While these responses were step functions of desiccation, we found that in deterministically assembled active communities, respiration was lower and thermodynamic properties of organic matter were less favorable."? Just a suggestion, maybe the sentence is clear for other readers.
- *Edited (lines 24-26).*

7. Lines 38-42: Because of the crucial role that the framework by Hall et al. plays in this manuscript, I think the framework could be briefly described with a bit more detail here. It is somewhat difficult to keep track of what the Hall framework was prior to being modified here. Maybe a sentence clarifying that "Microbial membership influences community properties and microbial processes, which in turn regulate ecosystem fluxes; all of these components can be further modified by environmental variation (Hall et al. 2018)." This would also help readers who haven't read Hall et al. 2018.
- *Edited (line 46-48 and 49-53)*

8. Figure 1 is great, but is it missing a direct link from environment to function? Or is this implied by the combination of arrow 3 and 5?

● *This was on purpose and is implied by the combination of arrows 3 and 5.*

9. Lines 75-86: On the one hand, I understand the use of "emergent property" here because the relative influence of determinism and stochasticity is unpredictable on the basis of membership alone. But on the other hand, the relative contribution of stochastic and deterministic processes is a tally of the processes that shape microbial community membership, which could be said to emerge from the assembly processes. So, I suppose I'm getting turned around here about what is emergent: microbial community structure or the ecological processes that generate structure from which they were inferred? I'm not suggesting a change to this terminology, I'm just noting my conflicting thoughts about it and maybe looking for some clarification in the text on why the processes are an emergent property of the community, instead of vice versa.

● *A very interesting point, thank you for raising it. We agree, it is a little fuzzy in terms of what comes first and what 'counts' as an emergent property. Taking inspiration from Hall et al. (2018) it seems that the quantitation of assembly is an emergent property, per their definition of any property: "Microbial community properties (Fig. 2) represent an integrated characteristic of the microbiome that has the potential to predict or at least constrain the estimates of microbial processes." They go on to state "It is generally agreed that emergent properties refer to a quality of the whole that is unique and distinguishable from the additive properties of its constituents." As the reviewer noted, this suggests that the balance between stochastic and deterministic assembly fits squarely into the definition of an emergent property. However, we also see the reviewer's point that stochastic and deterministic factors lead to microbial membership. On the other hand, because assembly is influenced by biotic interactions, membership itself influences assembly. Given all of that, we elected to add a few sentences about this in the Introduction (lines 97-100) and modify the conceptual model to have a double headed arrow between microbial membership and microbial properties. We feel this might often be a reflection of reality, whereby other kinds of emergent properties (e.g., biomass density) may feedback to influence membership.*

10. Line 344-360: I'm not sure I fully understand what to do with the CPI output. The writing here is clear. I get the concept, I get the computation of the metric, and I get the interpretation of the metric in terms of the relative contributions of each bioreactor. But I'm kind of missing an explicit statement of what the CPI values can tell us about microbial processes in temporally varying environments in this study. It tells us about the variation among replicates, which should, in principle, be rather low if the sediment community was well homogenized. But what would it mean

(ecologically, in the context of the conceptual framework) if a CPI value was close to 0.5 or 1? Maybe a hypothetical example or some hypotheses relating desiccation frequency to CPI in the context of Fig. 1 (arrow 5) would be helpful.

- *In the context of this kind of experiment we think of the treatments as potentially changing the probability of outliers (i.e., hotspots) emerging. That is, we hypothesize the frequency of wet/dry transitions and/or the duration of drying could lead to different levels of variation (e.g., in microbial communities) across replicates. For example, each time sediments go dry, the exact spatial distribution of water films that remain will differ across the replicates. In turn, it seems plausible that greater among-replicate variation would be expected in treatments with more wet/dry transitions. High values of CPI require variation across interrogated locations/times (in our case, treatment replicates) because high CPI results from large contributions from hot spots/moments. In turn, a hypothesis is that CPI will increase with the number of wet/dry transitions. That hypothesis was rejected in this experiment, though we feel it deserves additional evaluation, potentially in studies with larger sample sizes per treatment to more robustly quantify CPI. Text was added to the manuscript along these lines, immediately prior to the Results section (lines 397-415).*

11. Lines 361-429: I appreciate the extraordinarily clear presentation of the results. It was a joy to read. Lots of interesting aspects of this study to weave together and it was very expertly done here by the authors.
- *Thank you.*

12. Lines 452-453: I think this is an important point that could be emphasized more strongly. Why is this particular aspect of the study so crucial? That many microbes in the environment are inactive, and that active microbes remain sensitive to abiotic stresses while also being responsible for ecosystem functioning seems like a key detail to highlight in the context of the framework.
- *Edited (lines 514-516 and 521-527).*

13. Lines 469-473: One reference that might be helpful for the discussion is Chase (2007) Drought mediates the importance of stochastic community assembly, PNAS.
- *Added.*

14. Lines 482-489: Great point. Yes, I think the environmental effects are strong drivers of the relationships observed here. There could also be a biomass effect if the desiccated treatments simply have fewer cells, most of which may be inactive due to desiccation stress.

- *Agreed. Thank you*

15. Lines 517-520: This is a really cool aspect of the study! Interesting way to show the link between desiccation in the environment and thermodynamic favorability for microbial growth.
- *Thank you*

16. Lines 539-544: One thought about the CPI output here is that the difference between the 31- and 34-day treatments is actually greater than their nearby position on the continuous x-axis might imply. The 34-day treatment experienced more or less constant conditions, and was only rewetted during the incubation. The 31-day treatment experienced a single pulse of water followed by a subsequent redrying event prior to the incubation. Perhaps this pulse generated among-replicate variation in the microbial community such that the reactor with high function was also very different in community composition. While homogeneous selection dominated this treatment overall, if the null model was constructed using all taxa observed in the study, it's not surprising only a subset of these taxa was able to survive desiccation (hence low beta-NTI). But if different taxa survived in different replicates, this could explain the high CPI. Is raw beta-diversity (or a within-treatment beta dispersion metric) of the active *community a better predictor of CPI than the beta-NTI metric for this study?*
- *To address this we regressed CPI against beta-dispersion for the putatively active and whole communities. We did not find a significant relationship between raw beta-diversity and CPI for either whole ($R2 = 0.003$, $p = 0.92$) or putatively-active communities ($R2 = 0.041$, $p = 0.70$). We included this information in lines 488-490.*

17. Lines 556-558: Yes, very interesting hypothesis and I suspect this is likely to be the case as long as environmental stresses (like desiccation) are homogeneously distributed in space/time. If stresses are spatio-temporally asynchronous, you might find high CPIs as hot spots/moments shift in space and time in response to environmental forcing.
- *Thank you*

Technical corrections:

18. Line 79: Seems like there's a typo here. "is an emergent property" maybe?
- *Edited*

19. Line 107: "preferential use of OM"
   - *Edited*

20. Line 141: "biogeochemically"
   - *Edited*

21.Line 303: "reference framework graphic" should probably be "Figure 1"

   - *Edited*

22. Line 467: "outcomes"

   - *Edited*